# The Involvement of Ascorbic Acid in Cancer Treatment

**DOI:** 10.3390/molecules29102295

**Published:** 2024-05-13

**Authors:** Di Guo, Yuan Liao, Jintong Na, Liangliang Wu, Yao Yin, Zhengcheng Mi, Shixu Fang, Xiyu Liu, Yong Huang

**Affiliations:** State Key Laboratory of Targeting Oncology, National Center for International Research of Bio-Targeting Theranostics, Guangxi Key Laboratory of Bio-Targeting Theranostics, Collaborative Innovation Center for Targeting Tumor Diagnosis and Therapy, Guangxi Medical University, Nanning 530021, China; 202121459@sr.gxmu.edu.cn (D.G.); liaoyuan2024@163.com (Y.L.); najintong@sr.gxmu.edu.cn (J.N.); wuliangliang1211@126.com (L.W.); yinyao@sr.gxmu.edu.cn (Y.Y.); mzc63612111@163.com (Z.M.); 18934402291@163.com (S.F.)

**Keywords:** vitamin C, tumor, delivery, cancer treatment

## Abstract

Vitamin C (VC), also known as ascorbic acid, plays a crucial role as a water-soluble nutrient within the human body, contributing to a variety of metabolic processes. Research findings suggest that increased doses of VC demonstrate potential anti-tumor capabilities. This review delves into the mechanisms of VC absorption and its implications for cancer management. Building upon these foundational insights, we explore modern delivery systems for VC, evaluating its use in diverse cancer treatment methods. These include starvation therapy, chemodynamic therapy (CDT), photothermal/photodynamic therapy (PTT/PDT), electrothermal therapy, immunotherapy, cellular reprogramming, chemotherapy, radiotherapy, and various combination therapies.

## 1. Introduction

Vitamin C (VC), scientifically known as ascorbic acid, is a vital water-soluble nutrient predominantly found in a variety of fruits and vegetables, including red peppers, kiwifruits, strawberries, oranges, tomatoes, and broccoli [1]. Within the human body, it participates in intricate metabolic processes, notably contributing to collagen synthesis, which is essential for the integrity and functionality of the skin, blood vessels, and connective tissues [2]. Additionally, it supports immune function [3], aids in neurotransmitter synthesis [4], facilitates iron absorption to prevent anemia [5], accelerates wound healing [6], promotes cardiovascular health [7], regulates fat metabolism [8], provides neuroprotection [9], and preserves bone integrity [7]. As a potent antioxidant, the hydroxyl group on VC’s lactone ring double bond plays a crucial role as a donor of protons and electrons, effectively reducing reactive oxygen species (ROS), such as superoxide anions, hydroxyl radicals, and singlet oxygen [10]. Exogenous VC supplementation is widely used and renowned for its protective effects against cancer development. Emerging evidence suggests that VC may exhibit pro-oxidant properties in its anticancer effects by acting as a reducing agent, facilitating interactions with metals like iron and copper, thus generating hydroxyl radicals to eliminate malignant cells [11]. VC exhibits dual properties: it is a potent antioxidant, effectively reducing the generation of oxygen free radicals, yet it may also exhibit pro-oxidant characteristics, causing cellular damage. Maintaining a balanced and moderate intake of VC is crucial for cellular health, providing protective effects while avoiding potential pro-oxidant risks.

In addition, VC is used as an adjuvant in cancer treatment. Observational studies suggest that higher intakes of VC, whether through diet or supplements, may correlate with a lower risk of gastrointestinal cancers [12]. Laboratory studies further reveal VC’s capabilities to induce apoptosis and inhibit proliferation in cancer cells, affecting key cellular signaling pathways that regulate growth and division [13]. VC has been observed to boost the effectiveness of some chemotherapy drugs and radiation therapy, enhancing their impact on cancer treatment [14].

This review examines the diverse applications of VC in various cancer treatment modalities, including emerging methods, such as starvation therapy, chemodynamic therapy (CDT), photothermal/photodynamic therapy (PTT/PDT), electrothermal therapy, immunotherapy, cell reprogramming, conventional chemotherapy, radiotherapy, and their combinations. Additionally, advancements in drug delivery systems, particularly nanotechnology, have significantly enhanced the stability and targeted delivery of VC, thereby improving its therapeutic effects and reducing side effects. VC not only enhances the efficacy of cancer treatment but also influences treatment strategies through its biochemical properties, offering patients safer and more effective options. The primary aim of this review is to comprehensively evaluate and highlight the inherent potential value and applications of VC in modern cancer treatment methods.

## 2. Absorption, Transport, and Metabolism of VC

The absorption, transport, and metabolism of VC are crucial aspects for understanding its mechanisms in cancer treatment. In the human body, VC is primarily absorbed through the intestines. Its absorption is influenced by various factors, including dosage, dietary habits, and individual differences [15]. VC mainly exists in the reduced state in the blood and is transported to various tissues and organs through the bloodstream, participating in various physiological processes [16]. Its metabolic pathways include reduction and oxidation reactions, forming metabolites such as dehydroascorbic acid [17].

The consumption of VC is typically achieved through three primary methods: daily dietary intake, dietary supplements, and VC medications. Clinical studies indicate that synthetic VC in supplements and medications exhibits comparable bioavailability to natural VC [18]. All three approaches effectively supplement VC levels in the body. Recent research suggests that, to maintain optimal immune system function, VC intake should be increased to 200 milligrams per day [19]. Individuals with higher body weights require an additional 10 milligrams of VC for every 10 kg of body weight to attain similar plasma concentrations as those with lower body weights [20]. The body tightly regulates the concentration of VC in plasma and tissues through processes, such as intestinal absorption, renal reabsorption, and excretion. The plasma VC concentration is rigorously controlled, with 100% bioavailability from a single oral dose of 200 mg of VC. Bioavailability significantly decreases beyond a dosage of 500 mg [21]. Recent studies indicate that intravenous injection can achieve high VC doses in patients, resulting in a more significant increase in plasma VC concentration compared to oral administration [22]. VC is absorbed into the blood through the small intestine and is regulated by the kidneys through the balance of reabsorption and excretion, with unreabsorbed VC excreted in urine (Figure 1A). Ascorbic acid (AA) and its oxidized form, dehydroascorbic acid (DHA), are the predominant reduced and oxidized forms of VC, respectively [23].

VC enters cells via sodium-dependent VC transporters (SVCTs) and glucose transporters (GLUT1). Upon cellular entry, DHA undergoes reduction to VC facilitated by glutathione (GSH), converting to oxidized glutathione (GSSG). Subsequently, triphosphopyridine nucleotide (NADPH) reduces GSSG back to GSH [24]. AA is transported by SVCT1 and SVCT2, sodium-dependent vitamin transporters present in various tissues [16]. SVCT1, predominant in the liver, lungs, kidneys, intestines, and skin, maintains systemic homeostasis, while SVCT2 facilitates tissue-specific absorption, crucial for growth and development [25,26]. The body regulates ascorbic acid levels via negative feedback. Elevated intracellular concentrations downregulate SVCT transporter expression on intestinal cell surfaces [27]. DHA enters cells through GLUT1, a member of the SLC2 protein family primarily transporting glucose. The facilitated diffusion of DHA via GLUT1 is modulated by glucose concentration and cellular demands, especially in colon and lung cancers [15]. Colon and lung cancer cells, which require increased glucose for proliferation, enhance DHA uptake by upregulating GLUT transporters. Notably, cancer cells with Kirsten ratsarcoma viral oncogene homolog (KRAS) or B-Raf Proto-Oncogene (BRAF) mutations show enhanced DHA absorption. Their heightened GLUT1 expression accelerates DHA to VC conversion, depleting GSH and NADPH and elevating ROS levels (Figure 1B) [28].

## 3. Oral VC for Cancer Prevention

VC is widely acknowledged as a prevalent antioxidant present in fruits and vegetables. It is utilized as a dietary supplement to counteract oxidative stress-induced chronic ailments, such as cancer, cardiovascular diseases [7], hypertension, diabetes [29], Alzheimer’s disease [30], and infections [31], among others. While some research indicates that VC supplements do not significantly affect cancer risk, others propose a site-specific effect of VC, which might hinder certain malignant tumors without impacting others. To evaluate the potential causal influence of dietary antioxidant VC consumption on cancer, Shinjini Ganguly et al. discovered that oral VC supplementation (30 mg/kg/day in guinea pigs) demonstrated a capacity to inhibit oxidative damage and apoptosis, thus effectively preventing tumor development [32]. D Zhang et al. illustrated through observational studies that a high dietary intake of VC is inversely related to the risk of cancers like esophageal and bladder cancer [33]. However, research by Susanna C. Larsson et al. suggests that circulating VC is genetically associated with specific malignant tumors but not with esophageal, gastric, or pancreatic cancer, potentially contributing to the prevention of small intestinal and colorectal cancers [34]. Additionally, studies indicate that apart from VC, the combined action of various compounds in vegetables and fruits, such as phenolic phytochemicals, vitamins, dietary fibers, indoles, allium compounds, and selenium, may exert a more significant impact on cancer prevention [35]. Future case–control or prospective cohort studies should be meticulously designed and controlled to explore the effects of diverse vitamin supplements on carcinogenic risks, considering the diverse mechanisms, which may or may not be associated with antioxidant properties.

## 4. High-Dose Intravenous Administration of VC in Cancer Therapy

The administration of high doses of VC intravenously has garnered significant attention within the realm of cancer therapy. Linus Pauling postulated that the high doses of VC could provide a theoretical basis for combating cancer due to its antioxidative properties. Nobel laureate Pauling and his collaborators observed that administering high doses of VC could potentially extend the overall survival and alleviate symptoms in patients with advanced cancer [36]. Animal studies conducted by Guoping Wang et al. demonstrated that daily high-dose intravenous administration of VC notably decreased tumor volume [37]. Hoffer et al. conducted a Phase I-II clinical trial, where high-dose IVC was combined with cytotoxic chemotherapy in patients with advanced cancer. The study aimed to evaluate the safety, tolerability, pharmacokinetics, and efficacy of IVC. Doses ranged up to 1.5 g per kilogram of body weight per infusion. This study confirmed that these dosages were well tolerated and safe under clinical conditions, showing only transient adverse effects, which were manageable [38]. VC demonstrates antioxidative effects at physiological plasma concentrations but exhibits pro-oxidative effects at high concentrations [39]. Elevated levels of VC can elevate the levels of reactive oxygen species, thereby exerting anti-tumor effects. Moreover, high-dose VC has been shown to selectively eliminate colorectal cancer cells harboring the oncogenes KRAS or BRAF activating mutations [40]. Furthermore, metastatic tumor cells can adapt to the oxidative environment of the bloodstream by enhancing their antioxidant defense mechanisms [41]. Both biological and preclinical studies indicate that high-dose intravenous administration of VC, when combined with conventional chemotherapy agents, can synergistically enhance the efficacy of cancer therapy [42]. For individuals without cancer, large doses of intravenous VC are generally considered safe, with thirst and increased urinary output being the primary side effects [39]. Despite some clinical trial findings suggesting the potential anti-tumor effects of VC in certain tumor types, its precise mechanism of action remains unclear. Future research should focus on elucidating the mechanisms of VC and its synergistic effects with other treatment modalities to optimize cancer therapy outcomes.

## 5. The Mechanism of VC on Tumors

VC, a potent antioxidant, has an unknown molecular mechanism for inhibiting tumor occurrence. However, its anticancer mechanism has been confirmed to primarily involve neutralizing and eliminating free radicals. Free radicals are highly active molecules generated due to cellular metabolic activities. They have the ability to damage cell structures and functions, thereby promoting cancer onset and development. VC stabilizes free radicals by donating electrons, preventing them from causing harm to cellular tissues [43]. Additionally, VC can enhance intracellular antioxidant capacity by synergizing with other antioxidants such as vitamin E [44]. The redox state within cells is crucial for maintaining normal physiological functions, with tumor cells often exhibiting a different redox state compared to normal cells, tending to be in a more reduced state. Furthermore, VC interacts with intracellular transition metal Fe^2+^ ions (unstable iron), generating iron ions (Fe^3+^) through the Fenton reaction. This interaction produces a large amount of ROS, ultimately leading to cancer cell death [36] (Figure 2).

Influencing gene expression, VC regulates cell growth and differentiation, contributing to its anticancer effects. It impacts deoxyribonucleic acid (DNA) methylation [45], Ten-eleven translocation family protein (TET) enzymatic activity [46], hypoxia-inducible factor-1α (HIF-1α) signaling [15], and transcription factors (p53, NF-κB, AP-1) [47,48,49] modulate the expression of vascular endothelial growth factor (VEGF) [50], influencing tumor suppressor genes and oncogenes. This intricate regulation extends to key processes, like the cell cycle, apoptosis, and proliferation, effectively controlling cancer cell growth and spread. Beyond its direct effects on cancer cells, VC is recognized for enhancing immune cell activity, particularly lymphocytes and macrophages. This improvement enhances their capacity for identifying and eliminating cancer cells, leading to heightened immune cell proliferation, migration, and cytotoxicity. As a result, the body’s resilience against tumors is reinforced. Furthermore, VC participates in the regulation of inflammatory reactions, influencing the secretion profiles of tumor-associated cells and constituents of the extracellular matrix. It aids in adjusting the tumor microenvironment by controlling acid–base equilibrium, oxygen levels, and nutrient provision [51]. These varied mechanisms together contribute to the extensive anti-tumor impact of VC.

## 6. The Application of VC Delivery System in the Treatment of Cancer

It is well known that ascorbic acid is susceptible to the influence of heat, light, moisture, oxygen, and metal ions. Researchers have employed various encapsulation techniques, such as complex coacervation, spray drying, emulsification, and liposomes, to enhance the performance of VC. Anjali Khuntia et al. used different concentrations of gelatin and pectin as wall materials, combined with different concentrations of VC, and found that different pH values and temperatures affect the release of VC [52]. Anh Dao Thi Phan et al. utilized spray drying technology with maltodextrin as the carrier material for VC, increasing the stability and encapsulation efficiency of the juice powder [53]. Wanping Zhang’s study found that P/O/W multiple emulsions demonstrate superior encapsulation stability when contrasted with W/O/W multiple emulsions. Following a 2-week period, the encapsulation rate remains above 75%, whereas W/O/W multiple emulsions maintain levels below 60% [54]. Microencapsulation of VC in casein gel prepared VC capsules, namely micro-cheese powder, which significantly improved the stability of VC and delayed degradation [55]. Anjali Khuntia et al. prepared nano-liposomes containing VC through ultrasonication and optimized wall material proportions using thin-layer dispersion technology, exhibiting the highest encapsulation efficiency (94.18%) and storage stability, prolonging its lifecycle in the body [56].

The manipulation of carrier structure and properties can achieve precise control over the release rate of VC, thereby enhancing its effectiveness in drug delivery systems. Aldo Ugolotti and colleagues investigated the influence of geometric factors, such as curvature, thickness, diameter, and rolling direction, of nanotubes on the adsorption of VC by different structured titanium dioxide nanotubes and optimized the utilization performance of VC in nano-carriers [57]. Furthermore, through surface modification or functionalization, nano-carriers can achieve control over the release and targeted delivery of VC, specifically binding to lesion tissues or cells. Elnaz Tehrani and colleagues fabricated chitosan microspheres encapsulating VC through crosslinking using two agents: sodium tripolyphosphate and Tween 80. This method aimed to attain controlled release of VC [58]. Yao Peng and colleagues designed and synthesized brain-targeted glucose-VC (Glu-Vc) derivatives, showing excellent targeting ability of Glu-Vc-modified liposomes in in vivo evaluation [59]. Maryam Ghanbari-Movahed and collaborators enhanced VC receptor involvement by functionalizing Au-AA-DAPT NPs. This augmentation resulted in an increased cellular uptake of DAPT by cancer stem cells (CSCs). Consequently, the targeting capability of Au-AA-DAPT NPs to CSCs is improved, leading to the inhibition of Notch activity in breast cancer stem cells [60]. The introduction of derivative VC compound nanotechnology is also considered to overcome the instability of VC. Okan Icten et al. synthesized magnetic nano-composite materials using derivative VC compound nanotechnology, containing single or double complexes of the elements boron and VC, for targeted delivery and therapeutic applications [61].

Additionally, Pejman Shahrokhi and Mohamed Korany et al. validated 99mTc-VitC as a promising radiopharmaceutical for solid tumor imaging. This compound is also employed in single-photon emission computed tomography/computed tomography (SPECT/CT) imaging, facilitating early tumor detection through radiolabeled vitamin administration. This advancement opens avenues for enhanced therapeutic outcomes and diminished toxicity [62,63].

In terms of gene regulation, the combination of exogenous VC and nanoparticle-mediated transfer of the wt-101F6 gene can promote the uptake of VC within cells and accelerate the formation of cytotoxic hydrogen peroxide (H_2_O_2_), thereby effectively inhibiting the growth of tumor cells through non-caspase-dependent pathways of cell apoptosis and autophagic cell death [64]. The joint application of VC encapsulation and nano-carriers in tumor therapy has shown significant anti-tumor effects and improved survival rates (Table 1).

VC still faces challenges in biological distribution, drug delivery, and persistence in therapy. Consideration should be given to the rate of metabolism and degradation of VC in the body, as well as its differences in various types of cancer, to provide a deeper understanding for optimizing treatment regimens.

## 7. VC Enhances the Efficacy of Starvation Therapy

Cancer starvation therapy, a strategy that entails obstructing blood supply and depriving tumors of glucose and essential nutrients, has garnered considerable attention as a promising cancer treatment approach. Glucose acts as the primary fuel for tumor cells, with the oxidized form of VC, DHA, being transported via the glucose transporter protein GLUT1. Tumor cells undergo a metabolic shift from oxidative phosphorylation to glycolysis to meet their energy demands. Consequently, an excess of VC may impede glucose transport and adenosine triphosphate (ATP) production, triggering an energy crisis and eventual cell death (Figure 3) [65]. Furthermore, GLUT1 expression indirectly influences the effectiveness of tumor treatment. Research suggests that mutations in the downstream mitogen-activated protein kinase (MAPK) pathway activated by KRAS or BRAF result in the upregulation of GLUT1 expression, thereby enhancing DHA uptake and non-selectively eliminating gastric cancer and carcinoma of the colon. This, in turn, further affects glycolysis and intracellular redox balance [66].

The relationship between VC and other nutrients currently lacks direct evidence. In breast cancer tumors, hypoxia-induced HIF-1 stabilization upregulates glycolysis, angiogenesis, tissue remodeling, and metabolic stress due to rapid cell growth caused by hypoxia. However, a negative correlation between VC content and HIF activation has been observed in different tumor tissues. VC inhibits HIF activation, thereby suppressing glycolysis, angiogenesis, and inducing energy crisis and cell death [67]. Additionally, research indicates that elevated doses of VC disrupt glycolysis. They can be combined with the anti-diabetic medication metformin, which also diminishes tumor (acute myelocytic leukemia and other solid tumors) burden through mitochondrial complex I inhibition [68].

A review of recent clinical studies and case analyses suggests that VC, as a co-factor, holds certain potential application value in starvation therapy (Table 2).

## 8. The Application and Impact of VC in Chemodynamic Therapy for Cancer

Chemical Dynamic Therapy (CDT) represents a novel tumor treatment approach rooted in the dynamic nature of chemical reactions. CDT utilizes chemical reactions to generate toxic reactive substances within tumor tissues. These substances can induce oxidative reactions within cells, causing damage to cellular components such as nucleic acids and proteins, ultimately perturbing cellular homeostasis and potentially triggering apoptosis.

In CDT, commonly employed chemical reactions include the Fenton reaction and the Haber–Weiss reaction. The Fenton reaction is a chemical process that produces hydroxyl radicals (•OH), typically catalyzed by H_2_O_2_ and transition metal ions, such as iron ions [69]. On the other hand, the Haber–Weiss reaction complements the Fenton reaction. Through redox reactions, it converts hydrogen peroxide into superoxide anions and hydroxyl radicals, further enhancing the production of ROS (Figure 3).

In the realm of VC, historical debates swirl around its antioxidant and pro-oxidant facets. While VC’s antioxidant prowess stands firmly grounded in established knowledge, support for its pro-oxidant nature remains inconclusive. This uncertainty stems from the nuanced conditions required for VC to exhibit pro-oxidant tendencies, such as elevated concentrations and the presence of catalytic metals [70]. Recent investigations propose that VC’s pro-oxidant potential undergoes a substantial boost when formulated in nanostructures or combined with nanoparticles. To illustrate, for oral squamous cell carcinoma, inventive minds have crafted a nano-microneedle patch, housing a blend of VC and iron ions. This concoction, when introduced into oral squamous cell carcinoma cells, triggers the Fenton reaction by leveraging the oxidative iron (Fe^3+^) and VC present within the cells. The resultant hydroxyl radicals induce apoptosis in tumor cells, marking a precise strategy for treating malignancies [71].

In recent times, the focus has shifted towards the Fenton reaction-derived ROS in cancer treatment, attributed to limited catalase, elevated redox iron, and the alkaline pH in lysosomes. Nevertheless, the integrity of lysosomal membranes is safeguarded by heavily glycosylated proteins, with various mechanisms in place to promptly mend any damages. To achieve substantial ROS generation and induce complete lysosomal membrane disruption, scientists have engineered a lysosome-targeting ROS stimulant named N-(3-aminopropyl) methylimidazole-grafted cross-linked ascorbic acid vesicles (VC@^N3AM^cLAVs). Both laboratory and animal studies have demonstrated the efficacy of VC@^N3AM^cLAVs in boosting ROS levels and effectively converting H_2_O_2_ into •OH radicals. Ultimately, activation of the iron death pathway results in irreversible mouse colon cancer cell demise, suggesting VC@^N3AM^cLAVs as a potential candidate for anticancer therapy [72].

In the realm of CDT, emphasis is placed on the choice of suitable chemical reaction conditions and drug formulations. CDT presents certain benefits over conventional tumor treatments, including controlled therapeutic outcomes, reduced side effects, and targeted action on tumor cells. VC plays a pivotal role in facilitating ROS production as an antioxidant. Moreover, converting VC into nanoparticles can boost its bioavailability and antioxidant properties, thereby enhancing its efficacy in tumor therapy (Table 2).

## 9. Application and Influence of VC in Immunotherapy for Cancer

VC’s role in the immune system is significant. Primarily, it serves as a potent antioxidant, assisting in neutralizing free radicals in the body. This process reduces oxidative stress and supports immune system function. Supplementation with VC can elevate immunoglobulin levels, boost lysozyme activity, and enhance the antioxidant capacity level of *Oreochromis niloticus* L. Consequently, immune function strengthens, reducing infection risk [73]. Despite extensive research on VC’s antioxidant properties, its connection to immune regulation remains unclear. Some studies have shown that intravenous high-dose VC has anti-tumor effects in cancer mouse models (breast, colorectal, melanoma, and pancreatic tumor). These effects seem tied to immune system integrity, implying that VC’s anti-tumor activity relies more on its immune regulatory function than its pro-oxidant effects [74]. However, the correlation with the human immune system needs further verification. VC supplementation has diverse effects on the immune system, including increasing the activity and quantity of immune cells. Multiple studies have demonstrated that VC can modulate the function of various immune cells, such as Natural Killer Cells (NK cells), T lymphocytes (T cells), and dendritic cells. After supplementing with VC in peripheral blood mononuclear cell cultures, the number of mature NK cells significantly increases, and their function and phenotype improve (Figure 3). By stabilizing the expression of Forkhead box protein P3 (Foxp3), regulating TET activity, and enhancing Treg cell function, VC reinforces the anticancer function of immune cells [75]. VC positively influences the metabolic processes of immune cells. Studies suggest that VC can boost the metabolic activity of γδ T cells, including reducing ROS levels, increasing the proportion of cells in the G2/M phase, and enhancing glycolysis and oxidative respiration [76]. Moreover, VC facilitates antigen presentation and influences genes associated with immune responses. In the maturation of dendritic cells, VC pretreatment induces significant demethylation at nuclear factor kappa-B (NF-κB)/p65 binding sites. This, in turn, boosts the binding of signal transducer and activator of transcription 3(STAT3) to the Prdm1 promoter and downstream enhancers by promoting DNA demethylation. Consequently, it encourages plasma cell differentiation, bolstering the impact of immune cells on tumors [77,78]. The synergy of high-dose VC with immune checkpoint therapy, like anti- PD1 and anti- CTLA4, enhances the immune cells’ effect on tumors. This synergy leads to increased infiltration of CD4^+^ and CD8^+^ T cells and macrophages into the tumor microenvironment. It acts collaboratively with immune checkpoint inhibitors, elevating the treatment response across various cancers (lymphoma mouse model/mammary gland/large intestine/melanoma and pancreatic mice) [74,79]. Recent research has identified a substantial correlation between genetic predictions of VC levels and overall survival, tumor mutation burden, microsatellite instability, and the immune microenvironment in cancer patients (breast cancer, head and neck squamous cell carcinoma, renal clear cell carcinoma, and rectal adenocarcinoma). These findings offer potential insights for customizing VC intake based on genetic predictions to boost immune activity and improve cancer patient survival rates [80]. Nevertheless, despite existing studies indicating a role for VC in cancer treatment, additional research is required to deepen our understanding of its mechanisms of action and clinical applications (Table 2).

## 10. The Application and Impact of VC in Photothermal/Photodynamic Therapy and Electrothermal Therapy for Treating Cancer

The process of photodynamic therapy (PDT) and photothermal therapy (PTT) entails the exposure of materials with high efficiency in converting light into heat or generating ROS to a light source with specific wavelengths. Utilizing targeted recognition technology, these materials induce phototoxicity in specific cancer cells or diseased cells, achieving therapeutic effects. Iron chelators boost PDT sensitization, potentially destroying tumor cells by binding iron within cancer cells (mouse colon cancer cells and human gastric cancer cells) and forming active compounds. This process leads to the generation of reactive oxygen species, further augmented by VC [81]. In cancer photothermal therapy, VC is primarily used in conjunction with photothermal materials and nanoparticles for treatment. Through the rational carbonization of VC, researchers have developed water-soluble, biocompatible, and photoluminescent carbon nanodots, which have demonstrated promising effects in cancer (human astroglioma cell) photothermal therapy [82]. Yang and colleagues utilized starfruit juice abundant in VC and polyphenol antioxidants to produce gold nanoflowers via a green synthesis approach. These nanoflowers exhibit robust near-infrared absorption, making them well suited for photothermal therapy in a human breast cancer mouse model [83]. Similarly, Phan and team synthesized porous flower-shaped palladium nanoparticles using chitosan and VC through green synthesis. These nanoparticles demonstrate favorable biocompatibility and near-infrared absorption in breast cancer models, showcasing high efficacy in both photothermal therapy and photoacoustic imaging [84]. Modulated electro-hyperthermia (mEHT) involves using a 13.56 MHz radiofrequency current for regional hyperthermia [85]. Studies led by Junwen Ou et al. indicate that combining VC through intravenous injection with mEHT benefits the quality of life for non-small cell lung cancer (NSCLC) patients. This combination also extends both progression-free survival and overall survival [86]. In summary, VC acts as an adjuvant for photothermal and electrothermal therapies, enhancing their efficacy (Table 2). The effectiveness of VC in tumor treatment through Electrodynamic Therapy (EDT) remains to be substantiated.

## 11. The Application and Impact of VC in Cancer Cell Reprogramming Therapy

Extensive research has been conducted on the role of VC in cell reprogramming. Studies led by Esteban and colleagues demonstrated that the introduction of three or four factors (Oct4/Klf4/Sox2 or Oct4/Klf4/Sox2/cMyc) into mouse and human fibroblasts induces them to reprogram into induced pluripotent stem cells. VC was found to enhance the efficiency of this reprogramming process in both mouse and human fibroblasts [87].

Researchers have observed that VC has the potential to impede the epigenetic reprogramming of various cancer cells via the miR-302/367 cluster specific to embryonic stem cells. The inclusion of VC in the treatment of cells transfected with miR-302/367 could potentially hinder the expression of factors related to pluripotency while augmenting the tumorigenic properties of breast cancer cells. The adverse effects of VC on the reprogramming efficacy of the miR-302/367 cluster and its anticancer properties may be linked to the decreased expression of the TET1 gene when supplemented in the culture medium [88]. Consequently, considering the regulatory role of VC on TET1, it can be deduced that its application is not universally beneficial for reprogramming and may yield contrasting outcomes. Moreover, researchers have noted that the interplay between TET hydroxylases and VC influences the reprogramming of somatic cells, with their insufficiency potentially enhancing the reprogramming process (Table 2) [89].

## 12. Application and Impact of VC on Radiotherapy/Chemotherapy and Combined Cancer Treatment

In cancer treatment, VC holds significance. It contributes to enhancing patients’ quality of life and mitigating side effects like fatigue and nausea induced by chemotherapy or radiotherapy through its antioxidative properties [90]. Concurrently, there are numerous studies investigating the potential anticancer effects of VC as either a standalone therapy or an adjunct to conventional treatments. Research suggests that the combination of high-dose VC with specific anticancer medications can more effectively reduce cancer cell viability compared to using VC or chemotherapy alone [91]. Studies propose that VC, when administered alongside drugs, such as cisplatin, doxorubicin, and doxorubicin (an anthracycline), can enhance the anti-tumor activity of chemotherapy drugs, thus improving therapeutic outcomes and sometimes alleviating chemotherapy side effects [92,93,94]. Furthermore, the combination therapy of VC and cimetidine may hinder the progress of breast cancer in mice by inhibiting the production of mast cell mediators (histamine, VEGF, and TNF-α), reducing VEGF levels and restoring oxidative stress and inflammatory conditions to hinder tumor progression [95].

VC’s impact on radiotherapy is notable. Gamma rays, by generating free radicals upon interaction with biological materials, not only ionize cancer cells but also harm healthy ones. VC assumes a radioprotective role and might augment efficacy, contingent on its concentration and radiation dosage. H. Mozdarani et al. irradiated blood samples diluted in a complete RPMI-1640 culture medium with varying gamma ray doses (4, 8, and 12 Gy). At a low dose (10 μg/mL), VC displayed significant protection against radiation, mitigating radiation-induced human peripheral blood leukocyte apoptosis [96]. Concurrently, Prisyanto et al. discovered that VC and vitamin E (VE) intake could mitigate the decline in hemoglobin levels, white blood cells, and platelets induced by gamma ray exposure [97]. However, M. Konopacka et al. orally administered VC to mice for five consecutive days before or after irradiation with 2 Gy γ rays. The results indicated that the radioprotective effect of VC relied on its concentration (400 mg/kg/day) [98]. H. S. Taper et al., by pretreating mice with VC and K3 followed by a single exposure to 20, 30, or 40 Gy of X-rays, observed an enhanced therapeutic effect of radiotherapy on mice with solid transplantable tumors [99].

Furthermore, beyond its conventional application alongside standard chemotherapy and radiation treatments, VC finds utility in conjunction with various therapeutic modalities, such as chemodynamic therapy, photothermal therapy, immunotherapy, and nanomaterials, yielding encouraging outcomes. For instance, in a study by Fang Zhang et al., vanadium oxide (Vox) nano-enzymes and the photosensitizer dihydroxyphenylethanol (Ce6) were encapsulated within liposomes to create a tumor microenvironment (TME)-responsive nano-carrier named VC@Lipo. VOx nano-enzymes exhibit peroxidase-like activity, generating highly toxic hydroxyl radicals through Fenton-like reactions and the formation of ^1^O_2_, independently of light in the presence of H_2_O_2_. Meanwhile, Ce6’s photodynamic effect enhances the production of ^1^O_2_. Compared to individual treatment modalities, VC@Lipo demonstrates superior chemo-dynamic–photodynamic synergistic effects [100]. In another study, Yingying Zhang et al. devised a nano-system combining photothermal therapy and immunotherapy for a tumor treatment. They loaded indocyanine green (ICG) and brominated Sepeptronium bromide (YM155) onto mesoporous silica nanoparticles (MSNs) to fabricate MSN-ICG-YM155. Additionally, silica nanoparticles (nSiO_2_) coated with magnetic nanoparticles (MNPs) were coupled with anti-CD47 antibodies to form MNP@nSiO_2_-anti-CD47. MSN-ICG-YM155, functioning as a prodrug, effectively eradicates primary tumors, exposing tumor antigens for cancer immunotherapy. On the other hand, MNP@nSiO_2_-anti-CD47, acting as a post-drug, exerts robust anti-tumor immune effects on distant tumors (mouse melanoma). This synergistic approach effectively suppresses the growth of both primary and distant tumors [101].

In recent times, there has been broad interest in cancer research regarding CSCs and their involvement in various aspects, such as presence, proliferation, differentiation, migration, invasion, tumor initiation, progression, metastasis, drug resistance, and recurrence. While conventional treatments, like surgery, chemotherapy, and radiotherapy, are effective, they often overlook CSCs, leading to disease relapse [102]. Studies indicate that VC, when administered intravenously alongside chemotherapy drugs, can selectively target and eliminate tumor cells and CSCs. This effect stems from VC’s ability to regulate cellular redox status, influence epigenetic modifications, and participate in HIF-1α signaling [103]. Additionally, researchers have devised a 68Ga-citrate Positron Emission Computed Tomography (PET) imaging method to identify tumor types susceptible to VC by evaluating Transferrin (TF)/transferrin receptor (TFR) expression levels, thereby paving the way for personalized therapy [104].

In conclusion, these researchers have laid a solid foundation for conducting clinical trials of VC in cancer treatment through preclinical exploration. VC demonstrates potential anticancer activity in various cancer treatments, especially as part of combined therapies. It affects tumor cell proliferation, apoptosis, and the microenvironment through multiple mechanisms, providing new insights and approaches for cancer treatment.

## 13. Clinical Prospects of VC in Cancer Treatment

In numerous studies, VC has shown promise in hindering cancer cell proliferation in experimental models, yet conclusive clinical evidence on its effectiveness against cancer remains scant. Debates persist concerning the outcomes of high-dose intravenous VC, including questions about dosage, frequency, and anticancer impacts. Preliminary research suggests that administering VC through single intraperitoneal injections in mice yields inferior results compared to multiple injections. Frequent dosing seems to offer greater benefits, effectively suppressing HIF-1α and its downstream gene products more efficiently than alternate-day injections [105]. High-dose intravenous VC is generally considered safe, although it can induce severe adverse reactions in certain patients. Individuals lacking glucose-6-phosphate dehydrogenase (G-6-PD) are prone to intravascular hemolysis and should avoid gram-level VC doses [106]. Numerous clinical trials have demonstrated that, whether utilized alongside radiotherapy or chemotherapy agents, they elicit synergistic therapeutic effects, and patients tolerate high-dose intravenous VC well with minimal toxicity. In a Phase I clinical trial, intravenous high-dose VC (1.5 g/kg/day) combined with other anticancer medications (mFOLFOX6 or FOLFIRI) over a 14-day treatment period exhibited an objective response rate of 58.3% and a disease control rate of 95.8%, while also mitigating chemotherapy side effects and enhancing patient quality of life [107]. In general, the potential anticancer effects of intravenous and high-dose VC administration have been observed, with initial clinical trials indicating its safety and efficacy in eliminating tumor cells. Recent findings highlight VC’s diverse effects, encompassing anticancer properties, pro-oxidative cytotoxicity, and immune modulation. Moreover, high-dose VC demonstrates promise as a complementary cancer therapy, though conclusive evidence from clinical data and Phase III trials remains scarce. More extensive clinical studies are warranted to confirm the role of high-dose VC in cancer management. Understanding the mechanisms underlying high-dose VC’s actions can inform tailored treatment approaches for distinct patient cohorts and aid in identifying predictive biomarkers. Interindividual variations in VC metabolism and absorption result in divergent responses to its cancer-treating effects. Thus, personalized treatment strategies represent a crucial avenue for future investigation to comprehensively assess VC’s potential impact on cancer care. Furthermore, refining trial methodologies and employing comprehensive research approaches are essential for accurately assessing the efficacy of high-dose VC as an anticancer intervention. Given its non-toxic nature and affordability, high-dose VC merits further investigation and application in clinical settings.

## 14. Conclusions

VC exhibits considerable potential in cancer therapy through multiple mechanisms, including modifying the tumor’s metabolic environment, restricting access to crucial nutrients, and promoting cell death via oxidative stress. This review delves into VC’s varied applications in cancer treatment, such as chemodynamic therapy (CDT), photothermal therapy (PTT), starvation therapy, immunotherapy, and cellular reprogramming, as well as its use in conjunction with radiotherapy and chemotherapy. It highlights VC’s extensive and multifaceted role in enhancing the effectiveness of cancer treatments, its incorporation into novel therapeutic strategies, and underscores the critical need for further research to fully exploit its capabilities. Future research should further investigate VC’s interactions with immune regulation and its potential in cellular reprogramming. The effectiveness of VC treatment varies among individuals due to metabolic differences, underscoring the necessity for personalized treatment plans in clinical settings. Overall, integrating VC with other treatment modalities may enhance treatment efficacy, offering cancer patients more effective therapeutic options.

## Figures and Tables

**Figure 1 molecules-29-02295-f001:**
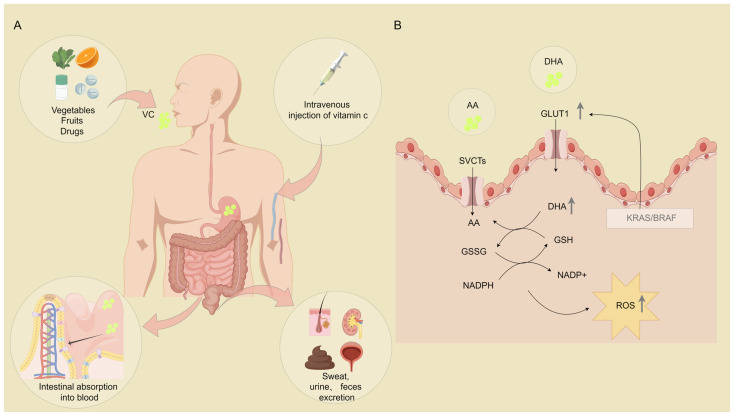
(**A**): Schematic diagram illustrating the absorption, transportation, and metabolism of VC in the human body. (**B**): Intravascular absorption mechanism of VC.

**Figure 2 molecules-29-02295-f002:**
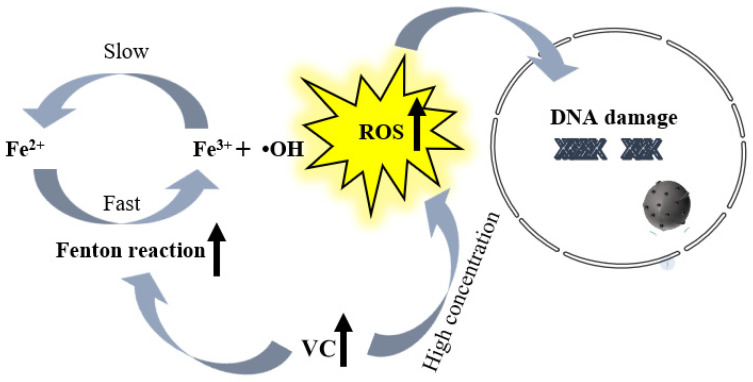
The process of cell death caused by Fenton reaction between VC and Fe^2+^.

**Figure 3 molecules-29-02295-f003:**
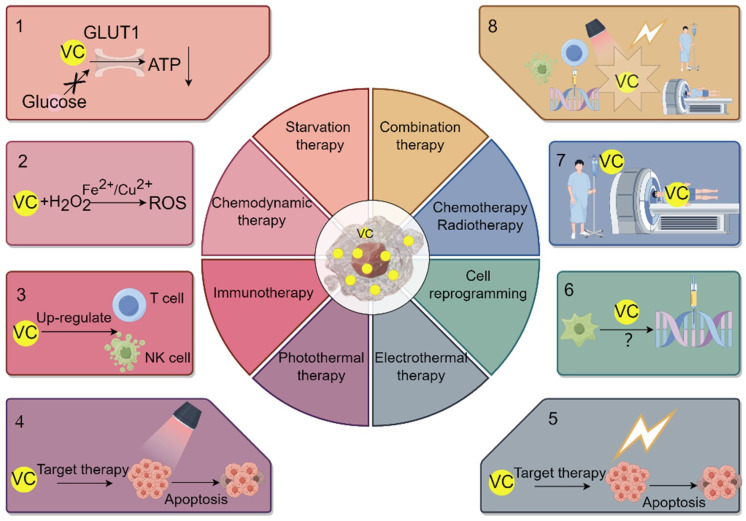
Schematic diagram illustrating the roles of VC in starvation therapy, chemical kinetics, immunotherapy, photothermal therapy, electrothermal therapy, cellular reprogramming, chemotherapy, radiotherapy, and combined therapy treatments. Abbreviations: GLUT1: Glucose transporters 1; ATP: Adenosine triphosphate; H_2_O_2_: Hydrogen peroxide; ROS: Reactive oxygen species; T cell: T lymphocyte; NK cell: Natural Killer Cell.

**Table 1 molecules-29-02295-t001:** Summary of VC delivery systems.

Category	Methods	Format	Function
Enveloping	Composite and cohesion	Different concentrations of gelatin and pectin are used as wall materials to form complex coacervates with different concentrations of VC.	The concentration of the wall material will affect the loading and release of VC.
	Spray drying	Maltodextrin serves as the carrier material for loading VC.	Enhancing the stability and encapsulation efficiency of fruit juice powder.
	Emulsification	P/O/W and W/O/W	The P/O/W system demonstrates superior encapsulation stability.
	Microcapsule granulation	VC capsules were prepared in casein gel	This capsule shows significant effectiveness in enhancing the stability of vitamins and delaying degradation.
	Nanoliposomes	Preparation of nanoliposomes containing VC.	It demonstrates a high encapsulation efficiency (94.18%) and storage stability.
Control of Carrier Structure and Properties	Modification of Titanium Dioxide Nanotube Carriers	Altering the geometric factors of titanium dioxide.	Enhancing the adsorption capacity of VC.
Surface Modification or Functionalization	Crosslinking Functionalization	Cross-linked preparation of VC-encapsulated chitosan microspheres.	To achieve control over the release of VC.
	Functionalized Targeting Capability	Synthesis of brain-targeted glucose- VC derivatives	To achieve control over the release of VC.
		Functionalization of Au-AA-DAPT NPs.	VC enhances the targeting ability of Au-AA-DAPT NPs towards CSC and suppresses Notch activity in breast cancer stem cells.
		Synthesis of magnetic nanocomposite materials containing single or double (ascorbic acid ester) complexes of boron and VC.	Used for targeted delivery and therapeutic applications.
	Contrast medium	99m Tc-VC	99m Tc-VC emerges as a potential radiopharmaceutical with high radiolabeling efficiency for solid tumor imaging using SPECT/CT.
Other	Gene regulation	The binding of exogenous VC with nanoparticle-mediated wt-101F6 gene transfer.	Facilitating intracellular uptake of VC.

Abbreviations: CSC: cancer stem cell; SPECT/CT: single-photon emission computed tomography/computed tomography.

**Table 2 molecules-29-02295-t002:** The role of VC in various treatment methods.

Category	Method	Function	Object
Starvation therapy	Inhibition of glycolysis	Excessive amounts of VC and mutations (KRAS or BRAF) activate the downstream MAPK pathway, restricting glucose transport and ATP production, leading to energy crisis and cell death.	Gastric cancers and carcinoma of colon
		VC inhibits the activation of HIF, thereby suppressing glycolysis, angiogenesis, etc., leading to energy crisis and cell death.	Breast cancer
		High doses of VC can impair glycolysis. When combined with the anti-diabetic drug metformin, it can alleviate tumor burden by inhibiting mitochondrial complex I.	Acute myelocytic leukemia and other solid tumors
Chemodynamic therapy	Enhancement of Fenton Reaction	Loading VC and iron ions onto nano-microneedle patches, utilizing intracellular high ferric oxide (Fe^3+^) and VC to undergo Fenton reaction, leading to death and apoptosis of oral squamous cell carcinoma cells.	Oral squamous cell carcinoma
		VC@^N3AM^cLAVs effectively enhances the generation of ROS, efficiently converting generated H_2_O_2_ into highly toxic •OH, initiating irreversible cell death of tumor cells through the iron death pathway.	Mouse colon cancer
immunotherapy	Reducing the risk of infection	Supplementing VC can elevate immunoglobulin levels, enhance lysozyme activity, thereby reducing the risk of infection.	*Oreochromis niloticus* L.
	Modulating immune cells	VC can regulate various immune cell functions, thereby strengthening the anticancer capabilities of the immune system.	breast, colorectal, melanoma and pancreatic tumor model
	Transcriptional regulation factor expression	VC increases the stability of human forkhead box protein Foxp3 expression, regulates TET activity and Treg cell function, thereby enhancing the anticancer capabilities of immune cells.	Human Treg cells
	Influencing relevant metabolism	VC can enhance the metabolic vitality of γδ T cells, increase the proportion of cells in the G2/M phase, and strengthen glycolysis and oxidative respiration.	γδ T cells
	Influencing relevant gene expression	Pre-treatment with VC leads to significant demethylation of NF-κB/p65 binding sites, enhancing the binding of STAT3 at the Prdm1 promoter and downstream enhancers, thereby promoting plasma cell differentiation and enhancing the immune cell’s efficacy against tumors.	Dendritic cells
	Collaborative immune checkpoint therapy	High-dose VC can synergize with immune checkpoint therapies, such as anti-PD1 and anti-CTLA4, and act on immune checkpoint inhibitors to enhance treatment response to a variety of cancers.	Lymphoma mouse model/mammary gland/large intestine/melanoma and pancreatic mice
	Gene prediction of VC intake	Individually adjusting VC intake through personalized gene prediction enhances immune activity and improves the survival rate of cancer patients.	Breast cancer, head and neck squamous cell carcinoma, renal clear cell carcinoma and rectal adenocarcinoma
photothermal/photodynamic therapy	Combined photosensitizer	VC and iron chelators (photosensitizers) enhance the generation of reactive oxygen species and the photothermal therapy to kill tumor cells. Iron chelators also enhance the efficacy of photodynamic therapy.	BALB/c mice model of mouse colon cancer and human gastric cancer cells
	Preparation of VC Nanocomposites	By rational carbonization of VC, water-soluble, biocompatible, and photo-luminescent carbon nanodots were obtained, making them suitable for photodynamic therapy treatment.	Human astroglioma cells
		Starfruit juice rich in VC and polyphenol antioxidants was utilized to prepare gold nanoflowers, exhibiting robust absorption in the near-infrared region, suitable for photothermal therapy.	Mouse model of human breast cancer
		Chitosan and VC were used for the green synthesis of porous flower-shaped palladium nanoparticles, exhibiting high efficiency in photothermal therapy and photoacoustic imaging.	Mouse model of Breast cancer
electrothermal therapy	Modulated Electrohyperthermia (mEHT)	Combining intravenous administration of VC with mEHT improved the quality of life for non-small cell lung cancer patients, extending both progression-free survival and overall survival.	Non-small cell lung cancer patients
cellular reprogramming	Reprogramming of mouse and human fibroblasts	In both mouse and human fibroblasts, the introduction of Oct4/Klf4/Sox2 or Oct4/Klf4/Sox2/cMyc transduction, combined with VC supplementation, significantly enhanced the reprogramming efficiency of both mouse and human fibroblasts.	Mouse and human fibroblasts
	Reprogramming of embryonic stem cell epigenetic regulation	VC counteracted the reprogramming of human breast cancer cells induced by miR-302/367, restoring their invasive and proliferative capabilities.	Human breast cancer cells
	The influence of TET hydroxylase on somatic cell reprogramming	The association between TET hydroxylases and VC influences the reprogramming of somatic cells. The deficiency of TET hydroxylases may enhance the reprogramming efficiency.	somatic cell
chemotherapy	Alleviating chemotherapy side effects	VC can enhance patients’ quality of life and alleviate side effects caused by chemotherapy through its antioxidative effects.	Pancreatic cancer/Cervical neoplasia/Renal cell carcinoma/Esophageal cancer/Prostate cancer patients
	Combining VC with Chemotherapy Drugs	The combined use of high-dose VC with certain anticancer drugs can more comprehensively reduce the viability of cancer cells compared to using VC or chemotherapy alone.	Beast cancer cells and gastric Cancer cells
	Influence of VC and Chemotherapy Drugs on Tumor Microenvironment	The combination therapy of VC and cimetidine can inhibit the production of mast cell mediators (histamine, VEGF, and TNF-α), reduce the levels of the VEGF as a marker of angiogenesis, and restore oxidative stress and inflammatory status to achieve tumor growth inhibition.	Mouse model of breast cancer
radiotherapy	Low-dose VC	Low-dose (10 μg/mL) VC exhibits protective effects against various doses of radiation, combating radiation-induced cell apoptosis.	Human peripheral blood leukocytes
	VC and VE	The intake of VC and VE was found to reduce the levels of hemoglobin, leukocytes, and platelet decline caused by exposure to gamma rays.	Blood cells and hemoglobin
	High concentration of VC	High concentrations of VC can enhance the therapeutic effects of radiation in mice (400 mg/kg/day).	Mouse of erythrocytes and leukocytes
	VC and K3	Pre-treating mice with VC and K3 was found to enhance the effectiveness of radiation therapy in mice with transplantable solid tumors.	Mouse with solid transplantable tumors
combination therapies	Chemodynamic therapy/Photodynamic effect	VOx nanoparticles generate highly toxic hydroxyl radicals ∙OH through Fenton-like reactions and the formation of ^1^O_2_. The photodynamic effect of Ce6 can also produce more ^1^O_2_.	Mouse breast cancer cell mouse model
	Photodynamic effect/Immunotherapy	MSN-ICG-YM155 as a prodrug exposes tumor antigens for cancer immunotherapy. MNP@nSiO_2_-anti-CD47 as a follow-up drug, in synergy with the prodrug, demonstrated potent anti-tumor immune effects on distant tumors.	Mouse skin melanoma cell model
Others	Targeted effect of epigenetic modification by VC	VC affects the ability of epigenetic modification, and when administered with chemotherapy drugs, intravenous VC at pharmacological doses can selectively kill tumor cells and target CSCs.	Cancer Stem Cells
	Individualized screening for VC-sensitive tumor types	68Ga-citrate PET imaging technology evaluates the expression levels of tumor TF/TFR, thus selecting tumor types more sensitive to VC for personalized therapy.	Mouse model of human prostate cancer

Abbreviations: KRAS: Kirsten ratsarcoma viral oncogene homolog; BRAF: B-Raf Proto-Oncogene; MAPK: Mitogen-activated protein kinase; HIF: Hypoxia-inducible factor; ROS: Oxygen species; H_2_O_2_: Hydrogen peroxide; •OH: Hydroxyl radicals; Foxp3: Forkhead box protein P3; TET: Ten-eleven translocation; STAT3: Signal transducer and activator of transcription 3; Oct4: Octamer-binding transcription factor; Klf4: Krüppel-like factor 4; Sox2: SRY-box transcription factor 2; VEGF: Vascular Endothelial Growth Factor; Vox: Vanadium oxide; MSN: Mesoporous silica nanoparticles; ICG: Indocyanine green; YM155: Bro-minated Sepeptronium bromide; MNP: Magnetic nanoparticles; CSC: Cancer Stem Cell; PET: Positron Emission Computed Tomography; TF/TFR: Transferrin/Transferrin receptor.

## Data Availability

Figure 1 and Figure 3, by figdraw.

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
