# Peer review of "The Involvement of Ascorbic Acid in Cancer Treatment"

_molecules, 2024, doi:10.3390/molecules29102295_

Round 1

Reviewer 1 Report

Comments and Suggestions for Authors

Vitamin C is an important nutrient in the human body that is involved in various essential metabolic processes. Recent research has shown that high doses of vitamin C may have anti-tumor effects. This article discusses how vitamin C is absorbed in the body and its potential for managing cancer. Vitamin C has shown promise in treating tumors at a molecular level and can affect tumor cells by altering their metabolic environment, limiting their access to nutrients, and causing oxidative stress. However, the effectiveness of vitamin C treatment may vary based on individual differences, so personalized treatment plans are necessary.

This review comprehensively reports on the use of vitamin C in the treatment of tumor diseases. The work is scientifically sound, well-organized, and thoroughly described. Various methods of delivering vitamin C for cancer treatment, such as starvation therapy, immunotherapy, and combination therapies, are also discussed. The references cited in this article seem appropriate and sufficient. As a minor revision, I suggest adding a scheme in paragraph “5 - The mechanism of vitamin C on tumors," to illustrate the important reactions mentioned in the sentences before ref. 25: "Furthermore, vitamin C interacts with unstable iron ions (Fe2+) in the cell, converting them to iron ions (Fe3+) through the Fenton reaction. This interaction produces a large amount of reactive oxygen species (ROS), ultimately leading to cancer cell death."

Reviewer 2 Report

Comments and Suggestions for Authors

Dear Authors,

The Review is about vitamin C supporting cancer treatment. I believe that the Review has a lot of potential, it requires necessary corrections before publication, without which it cannot be admitted to further publication steps:

Introduction - needs necessary expansion. It introduces the reader too poorly to the topic covered. Reading it, one can feel a great insufficiency. It would be worth adding information on which fruits and vegetables contain vitamin C. Of course, you can focus on those that have the most of it. 

In addition, what did the Authors mean when they wrote “complex metabolic processes”? which ones? specific examples can be given. 

I understand that antioxidant and pro-oxidant activity is important to the subject matter of the Manuscript, but other benefits and activities of vit. C cannot be overlooked in the introductory description.

In addition, a clear vit. C - cancer connection was missing. Has this effect already been proven in any studies? Yes, as the authors show later in the Review. However, it should have been pointed out here. 

Also missing at the end is a paragraph indicating what is covered in this Article, why the Authors undertook to write the Review acutely addressing this topic.

So the Introduction needs to be written from the beginning.

Chapter 2 - The information in the first paragraph should be supported with appropriate citations.

Fig. 1A - captions are illegible, please enlarge.

Chapter 3 - line 85 - you might want to add which cancers and support with appropriate citations.

lines 87-88 - again no specific information, please state which cancers you are referring to. 

The Manuscript lacks such a summary of what cancers vit. C is proven to work on (Table?).

Lines 112-113 - what does “high dose” mean? when is it high, and when is it toxic? After all, we know that an overdose of vit. C is detrimental to the body. This limit should be considered here.

Table 1 - “VC” instead of “Vc”

It is necessary to expand all abbreviations used in the Manuscript, e.g. DHA, GLUT1, MAPK, HIF....

Figure 2 - The caption for this figure needs to be corrected. Please include the abbreviation VC in the caption and add a legend with the expansion of all abbreviations used in it

Table 2 - please include the abbreviation VC in the information given in the Table. Under the Table, please expand all abbreviations used in it (in the form of a legend).

Line 288-289 and following - please use the abbreviation VC, since it has been introduced. Please standardize it. Reading the Review you can see that it is a conglomeration of several parts written by different Authors. Unfortunately, there was a lack of unification of it common version. Please work hard on this.

Conclusion - it is required to add the Authors' own opinion. Please include directions for future research. Here, the Authors' opinion on the topic taken up in the Review should be more strongly emphasized. 

In addition, the entire Manuscript contains numerous punctuation deficiencies (e.g., missing spaces) that need to be corrected. Its style does not comply with MDPI requirements (e.g., the font in Clonclusion).

Best regards

Round 2

Reviewer 2 Report

Comments and Suggestions for Authors

Dear Authors,

The corrections made by the Authors are satisfactory to me. I believe that the Article is suitable for publication.

In my opinion, explanations of abbreviations should be at the end of the Manuscript, and if the Authors have already decided on this form it should include all abbreviations, even those that are explained in the text.

Best regards